# Tools to Enumerate and Predict Distribution Patterns of Environmental *Vibrio vulnificus* and *Vibrio parahaemolyticus*

**DOI:** 10.3390/microorganisms11102502

**Published:** 2023-10-05

**Authors:** Lisa A. Waidner, Trupti V. Potdukhe

**Affiliations:** 1Hal Marcus College of Science and Engineering, University of West Florida, 11000 University Pkwy, Building 58, Room 108, Pensacola, FL 32514, USA; 2GEMS Program, College of Medicine, University of Illinois Chicago, 1853 W. Polk St., Chicago, IL 60612, USA; tpotdu2@uic.edu

**Keywords:** aquaculture, *Vibrio vulnificus*, *parahaemolyticus*, molecular, culture-dependent, culture-independent, environmental, oyster

## Abstract

*Vibrio vulnificus* (*Vv*) and *Vibrio parahaemolyticus* (*Vp*) are water- and foodborne bacteria that can cause several distinct human diseases, collectively called vibriosis. The success of oyster aquaculture is negatively impacted by high *Vibrio* abundances. Myriad environmental factors affect the distribution of pathogenic *Vibrio*, including temperature, salinity, eutrophication, extreme weather events, and plankton loads, including harmful algal blooms. In this paper, we synthesize the current understanding of ecological drivers of *Vv* and *Vp* and provide a summary of various tools used to enumerate *Vv* and *Vp* in a variety of environments and environmental samples. We also highlight the limitations and benefits of each of the measurement tools and propose example alternative tools for more specific enumeration of pathogenic *Vv* and *Vp*. Improvement of molecular methods can tighten better predictive models that are potentially important for mitigation in more controlled environments such as aquaculture.

## 1. Introduction

*Vibrio* are Gram-negative, flagellar bacteria that typically live in marine to estuarine waters [1]. Three major human pathogens in the genus include *Vibrio vulnificus*, *V. parahaemolyticus*, and *V. cholerae*, the first two of which are considered in this review. As facultative anaerobes, *Vibrio* metabolic demands can be met even in suboxic waters [2]. *Vibrio vulnificus* (henceforth abbreviated *Vv*) can enter open cuts or wounds via exposure to contaminated seawater and may cause necrotizing fasciitis (infamously known as “flesh-eating disease”). Systemic cases often necessitate surgery or peripheral amputations, and in rare cases, result in death [3,4]. *Vv* can also cause gastrointestinal disease through the ingestion of raw or undercooked shellfish and is the leading cause of reported acute gastroenteritis and septicemia cases due to foodborne illness [5]. Over 95% of all *Vv* cells associated with oysters are in the meat [6], and roughly 50% of *Vv* human infections are transmitted through food consumption [7]. Similarly, *V. parahaemolyticus* (*Vp*) is transmitted through the consumption of raw or undercooked shellfish, especially oysters [8]. Both *Vv* and *Vp* can exist in the water column in a planktonic state, but in response to certain molecular signaling events, can become sessile on surfaces such as shellfish, finfish, plankton, or suspended particulate material [9,10]. Although antibiotic-resistant *V. cholerae* resulting from the use of aquaculture probiotics may confer resistance to non-cholera *Vibrio* species, as reviewed elsewhere [11], numbers of isolates from environmental samples that are identified as *V. cholerae* can be quite low [12]. This review focuses on environmental and climate-related parameters known to be associated with higher incidences of *Vv* and *Vp* in several types of environmental samples.

Unfortunately, the global occurrence of outbreaks and rates of reported *Vv* and *Vp* cases may increase [13,14,15]. Vibriosis cases are predicted to rise, in part, due to anthropogenic climate change and associated extreme meteorological events and downstream changes to the waterbodies where *Vibrio* naturally reside [16,17,18,19]. Antibiotics usage in aquaculture areas may also lead to *Vibrio* populations with antibiotic tolerance (“persisters”, please see below). Additional considerations are the specific genetic and phenotypic properties of strains, serovars, and “serovariants”, which are extensive in both *Vv* and *Vp* species, as evident from just a single study of oyster-associated strains [20]. An increase in variant numbers among strains, i.e., increased genetic diversity, should be considered, especially when enumerating with functional genes, such as those that confer fitness in aquaculture facilities, fitness for sessile living state, and/or higher pathogenicity to humans [21]. Hence, it is particularly important to recognize the limitations and strengths of the various molecular tools for detection of *Vibrio* in environmental samples.

## 2. Objectives

The primary aims of this review are to (i) provide an overview of virulence genes used to call for and enumerate *Vv* and *Vp* in environmental samples, (ii) summarize current knowledge of environmental factors driving *Vv* and *Vp* success, and (iii) present examples of alternative molecular tools for ecological surveys and better predictions of *Vv* and *Vp* distribution changes with various environmental and climate-related perturbations.

## 3. Virulence Factors and Associated Marker Genes

### 3.1. Culture-Dependent and -Independent Method Considerations

Microbiologists often use culture-dependent methods. However, the limitation of first relying on growth in liquid or solid media before enumerating the bacteria may lead us to underestimate the numbers in the substrates of interest, such as fish tissue, sediment, and water column samples. The “viable but not culturable” (VBNC) state is a survival mechanism in which bacteria are alive, but do not grow in culture media. Entering the VBNC state often occurs in response to unfavorable conditions, such as, in the case of *Vv*, extreme salinity changes or large temperature drops [22]. “Persister” cells are those that are possibly induced by the usage of antibiotics in aquaculture areas, as previously reviewed elsewhere [23] Briefly, persisters are similar to VBNC cells, in that both types are more stress-tolerant than vegetative cells, but persisters are in a less dormant state than VBNC cells; additionally, persister cells are generally antibiotic-tolerant. Some surveys of the isolates of aquaculture samples resulted in significant number of isolates with multiple Antibiotic Resistance Genes (ARGs), especially from aquaculture waters where antibiotics usage was low [24]. A few ARG marker genes are provided in Table 1, and a more comprehensive list is available in [24]. For further review on *Vv* VBNC and persisters, please see [23], and for a proteomics study of resuscitation mechanisms of *Vp* from the VBNC state, see [25]. An extensive review of *Vibrio* VBNC and persister states in survival and epidemiological implications was recently published elsewhere [26]. While lower temperatures may lead to underestimation of environmental or mammalian–host-associated *Vv* [27,28], the opposite may be true for *Vp* in controlled culture conditions where salinity, not temperature, is the important factor affecting the VBNC state [29]. In slightly different culture conditions, increased temperatures can somewhat reverse the VBNC issue [30]. Additionally, physiological responses of oysters to hypoxia may influence the accumulation of *Vibrio* in the meat [31]. Other environmental stressors on the host organism, such as salinity, temperature, and metal concentrations may also influence *Vibrio* loads in the oyster [6,32].

A variety of methods have been reported for enumeration and screening for *Vibrio* in water, sediment and aquaculture samples. These include those based on solely culture-independent (“molecular”) tools, where DNA or RNA is extracted directly from the substrate and evaluated using deep sequencing of a marker gene (“amplicon” libraries), qPCR, RT-qPCR, metagenomics, or metatranscriptomics. Since the culture-dependent step is skipped, the bacterial VBNC state does not affect detection. In waterbodies annually experiencing wide temperature ranges, hence possibly inducing the VBNC state of *Vibrio*, considerations should be made when enumerating during cold months. For example, in the Chesapeake Bay, *Vv* could comprise ~10% to 50% of culturable bacterial populations in oysters [33], easily enumerated via direct-plating/colony hybridization (DP/CH; i.e., a method that is inherently culture-based). However, in the same estuary, *Vibrio* numbers declined with temperature, thereby evading detection with these standard culture methods [34].

Currently, most studies rely on a combination of enrichment followed by molecular verification. An example is MPN (most probable number)-qPCR (MPN-qPCR) or MPN-PCR, where the substrate is diluted in liquid culture, counts are based on the MPN method, and then turbid cultures are subjected to either non-quantitative PCR or quantitative PCR (qPCR). The non-quantitative (PCR) or quantitative (qPCR) step is designed to verify presence, or to obtain gene abundances from the species or strains of interest. MPN-PCR and MPN-qPCR have long been used for pathogen screening of environmental or food samples [35,36,37]. Other methods based on culture-dependent, followed by culture-independent tools include DP/CH and DP-colony PCR, both relying on the success of the *Vibrio* to first grow on a solid culture medium. All such methods, liquid- and solid-culture-based (DP/CH, DP/CH-followed by PCR confirmation, DP followed by colony PCR, MPN-PCR, and MPN-qPCR) rely on the “cultivability” of the species of interest. Thus, rather than using only conventional microbiological methods for detection and quantification of bacterial communities, it can be more valuable to use molecular methods that evaluate DNA or RNA for confirming the presence/absence or providing quantities of the strains of interest. Care must be taken to interpret the findings of any type of study (culture-dependent, or -independent) when modeling predictions of increased pathogenic *Vv* or *Vp* with any environmental or ecological factor [38]. A newer type of quantitative PCR (digital droplet PCR, ddPCR) has recently been used to quantify pathogenic *Vibrio* species in the environment [39]. Benefits of pathogenic screening bypassing the culture step, and performing qPCR or RT-qPCR directly on DNA or RNA extracted from the substrate, has been extensively reviewed elsewhere [40,41,42].

### 3.2. Marker Genes

Sequencing of conserved taxonomic marker genes, such as the 16S rRNA gene, 23S rRNA gene, and *rpoA*, can help in the taxonomic identification of various *Vibrio* isolates, regardless of species, strain, or pathogenicity status [43,44]. However, due to the highly conserved sequences of these phylogenetic markers, PCR of these genes are usually not suitable for screening for species or specific strains, and thus cannot be used to reliably differentiate between pathogenic and non-pathogenic strains. Genus-level molecular tools were used in earlier studies of environmental *Vibrio* [44,45,46], some of which are included in this review for describing the early findings of environmental parameters driving *Vibrio* success. “Functional” genes are now more often used as molecular markers to enumerate and/or score samples as positive or negative for *Vibrio.* Both *Vv* and *Vp* possess specialized genes that contribute to pathogen virulence by promoting phenotypes such as increased epithelial (intestinal) cell adhesion or erythrocyte (red blood cell) lysis (i.e., hemolysis). Some enteropathic bacteria are capable of host invasion through mucus-binding proteins, including those encoded by *gbpA* of *V. cholerae* [47] and *V. vulnificus* [48]. An early phenotypic screening tool for pathogenic *Vibrio* was the modified Elek test method [49]. A positive result, called the Kanagawa phenomenon, is characterized by the halo formation around colonies grown on Wagatsuma agar, signifying the colony contains a *Vp* strain actively expressing the ß-hemolysin. The Kanagawa phenomenon test must be validated with culture-independent methods, since pathogenic strains from human patients, fish and water that score *tdh*-positive do not always present the Kanagawa phenotype [50].

The *Vv*-specific virulence gene, *vvhA*, encodes an extracellular cytolysin (VVH) which, if present, often confers higher likelihood of pathogenesis, possibly by helping the *Vv* invade host intestines, thereby allowing for its entry into the bloodstream [51]. Published *vvhA* PCR primer pairs (Table 1) are routinely used to detect *Vv* in DNA from seawater, sediments, fish or shellfish tissues. Recently, *vvhA* has been used to enumerate total pathogenic *Vv*, while the virulence-correlated gene (*vcg*) is now considered as a stronger indicator of virulence in *Vv*. Both environmental (*vcgE*) and clinical (*vcgC*) genes are strongly associated with distinct types of the hemolysin gene, *vvhA*, among clinical and environmental isolates [52,53].

**Table 1 microorganisms-11-02502-t001:** Noteworthy *Vibrio* marker genes. Common marker genes that are used to identify and enumerate *Vibrio vulnificus* (*Vv*) and *V. parahaemolyticus* (*Vp*) in environmental samples are listed here. Although most studies rely on PCR to detect species-specific genes, selected genes from recent publications relying on transcriptomics, genomics or metagenomics are also presented.

Gene of Interest	Species		Encodes ^1^	Mode of Pathogenesis	PCR Primers ^2^	Reference ^3^
	*Vv*	*Vp*				
*vvhA*	✓		VVH cytolysin (hemolysin)	Extracellular cytotoxin, cytolytic pore-forming protein, pore-forming cholesterol-dependent cytolysin	vvhA_F/R	[34]
					vvhA-F/R	[54]
					vvhA 1973 rev/vvhA 1795	[55]
					F-vvh785/R-vvh990	[56,57]
*tdh*		✓	Thermostable direct hemolysin	Increase permeability of host cell membranes, act as toxic porins, result in cell lysis	tdh86F/tdh331R	[58]
*trh*		✓	*tdh*-related hemolysin		trh90F/500R	
*tlh*		✓	ThermoLabile hemolysin		tlh-F/R	[59]
*tdh*, *trh with IAC*^4^		✓	see above	see above	*tdh*, *trh with IAC*	[60]
*tdh*, *trh*, *tlh with IAC*^4^		✓			*tdh*, *trh*, *tlh with IAC*	[61]
*ToxR*	✓	✓	Regulatory transcriptional membrane protein	Strongly associated with the upregulation of the gene encoding *tdh* in *Vp* [62] or *vvhA* in *Vv* [63]	UToxF/vvtoxR (*Vv*); UToxF/vptoxR (*Vp*)	[64]
*vcgC*, *vcgE*	✓		“Virulence-correlated gene” (C, clinical; E, environmental)	Strongly associated with the expression of *vvhA*	vcgCP1/P3vcgEP2/P3	[52,53]
*vpm*		✓	*Vp* metalloprotease	Shows proteolytic activity towards type I collagen; degrades host tissue and may promote pathogen invasion	vpm1/2	[65]
*pilA*	✓		Type IV pilin	Involved in biofilm formation, adherence to human epithelial cells, and oyster colonization [66,67,68]	VvpAF3/VvpBR6	[69]
*pilF*	✓				PilF-F/R	[70,71]
*brpL*, *brpG*	✓		BrpL, BrpG	Involved in extracellular polysaccharide production, biofilm formation; affects rugose colony phenotype	N/A	[72]
*hupA*	✓		TonB-dependent heme and hemoglobin receptor	Important for iron acquisition during infections [73]	hupA_F1/R1	[74]
*sodB*	✓		Iron superoxide dismutase	Promotes survival and virulence under acid stress and phagocyte engulfment [75]	FeSOD_F1/R1	
*vvpE*	✓		ferric uptake regulator (Fur)	Regulation of *vvhA* expression	N/A	[76,77]
*rtxA*	✓	✓	Multifunctional-autoprocessing repeats-in-toxin (MARTX)	Improves antiphagocytosis, colonization, and dissemination to the bloodstream and other organs	N/A	[78,79]
*plpA*	✓		Toxic lipoprotein phospholipase A2	Causes necrotic cell death of epithelial cells, lyses human erythrocytes of membranes containing sufficient phosphatidylcholine	N/A	[80,81]
*wza*	✓		Polysaccharide export lipoprotein	Formation of capsule to allow bacterium to evade host immune system; increases *Vv* survival in the presence of serum	wza_F/R	[82]
*OmpU*	✓	✓	Conserved outer membrane protein, fibronectin binding protein	Toxic porin that forms nonspecific β-barrel channels allowing free diffusion of hydrophilic molecules across outer membrane; shown to exhibit resistance to bile and antibacterial peptides	OmpU R-F/R	[83]
*gbpA*	✓		GbpA, GlcNAc-binding protein A	Mucin-binding protein involved in colonization of host intestine	GBPA_qRT-F/R	[48]
Example regulatory genes used as PCR markers						
*HlyU*	✓	✓	DNA binding protein	Promotes activation of ExsA, a master regulator of type III secretion systems that control >40 genes, many of which are involved in virulence	NT398/399	[84,85]
*RseB*	✓		Periplasmic negative regulator of the alternative sigma factor E	Controls an extensive regulon involved in responding to cell envelope stresses; associated with the colony morphotype of extracellular polysaccharides	rseB5′USER/rseB3′USER	[86]
*rtxA*	✓		RTX toxin protein	Exported via Type I Secretion System, many roles in infection, including growth in host, and host cell necrosis and apoptosis	N/A	[87,88]
*vvpR*, *smcR*	✓		VvpR, SmcR	Homolog of *LuxR* gene in *V. fischeri*; quorum sensing master regulator, in *Vv*, regulates transcription of the maltose regulon	N/A	[87,89]
Example ARGs or ARG regulator genes used as PCR markers ^5^						
*blaCARB-17 like element*		✓	ß-lactamase	Intrinsic *Vp ampR*, disrupts beta-lactam ring present in beta-lactam-class antibiotics	CARB-VP-F/R	[90]
*tetB*		✓	tetracycline efflux MFS transporter	For active pumping of the tetracycline compound out of the cell	tetBDEFHJ-F, tetBD-R	[91]
*qnrS*	✓	✓	QnrS family quinolone resistance pentapeptide repeat protein	TetR/AcrR family transcriptional regulator associated with resistance to fluoroquinolones	qnrS-F/R	[24]

^1^ Name of protein encoded by gene of interest. ^2^ Published PCR primer pair name, name of primer used to call for gene of interest using molecular methods (N/A: PCR was not used, gene sequence groups identified via deep-sequencing or genomics/metagenomics tools). ^3^ Reference: study in which PCR primer pair was first developed and tested or employed to enumerate *Vv* or *Vp* in environmental samples. ^4^ IAC = Internal Amplification Control. ^5^ ARG = Antibiotic Resistance Gene. For ARGs, a more comprehensive list of PCR targets for screening of aquaculture samples is provided by [24].

The pilus has a major role in *Vibrio* pathogenesis. In pathogenic *Vv* strains, *pilF* and *pilA* encode major pilus structural subunits. Those, and potential regulator genes like *gacA* that may control *pil* expression, are involved in biofilm formation, oyster colonization, and adherence to host gut epithelial cells [66,67,68,92]. Common PCR primer pairs for enumerating the *pilA* and *pilF* genes show promise for seafood pathogen screening [69,70,71]. The expression of the toxic lipoprotein, *plp*, causes necrotic cell death of epithelial cells and lyses human erythrocytes of membranes. For example, *Vp* persisting in Pacific oyster tissues rely on types I and IV pili, as well as polar and lateral flagellar systems [93]. Not only has this study helped us to better understand the pilus mechanism at the *Vp*-oyster interface, but also provided a gateway for more in-depth studies, for evaluating potential effects of relevant environmental parameters, including joint salinity-pH condition. Expression of the aforementioned genes is directly involved in pathogenicity. However, important regulatory genes responsible for the expression of virulence genes are also used to assess potential pathogen loads (Table 1, regulatory gene examples). These include transcriptional activator *HlyU* [85], periplasmic regulator *RseB* [86], heme and hemoglobin receptor *hupA* [74], and quorum-sensing master regulator *SmcR* [76]. Another example is *BrpR*, a regulatory protein upstream of several genes known to be involved in *Vv* biofilm formation and rugose colony formation [72], indicative of the potential for pathogenicity (see Table 1, *brpL*, *brpG*).

Similarly, there are several virulence-associated genes in *Vp*. Generally, the hemolysin genes of major interest for screening environmental samples are *tdh*, *tlh*, and *trh.* Like VVH, *Vp* hemolysins increase the permeability of host cell membranes, act as toxic porins and could result in cell lysis [94]. The genes encoding the thermostable-direct hemolysin (*tdh*) and the thermostable-direct hemolysin-related hemolysin (*trh*) are the most commonly used *Vp* virulence gene markers (Table 1). The gene encoding the thermolabile hemolysin (*tlh*) is typically a species-specific marker that is used to call for and enumerate all *Vp*, regardless of pathogenicity. Not all strains of *Vp* are pathogenic, and samples that score *tdh-* or *trh*-positive (or both) are generally only those that are scored as “hemolysin-producing” [59]. Interestingly, synteny analyses of *Vp* hemolysin and nearby genes indicate gene order may confer differences in pathogenicity of strains isolated from human wounds or feces, water samples, and shellfish [95].

The ToxR protein, first described in *V. cholerae*, is a regulatory protein strongly associated with the upregulation of the gene encoding *tdh* in *Vp* [62] or *vvhA* in *Vv* [63]. The *toxR* gene can be used to identify pathogenic *Vp* and *Vv* in environmental studies [96,97], and a homolog is present in *V. alginolyticus*, another important related pathogen [64]. The *Vp* metalloprotease, encoded by *vpm*, promotes pathogen invasion via degradation of fish host tissue, especially type I collagen [65]; the *vpm* gene is another example of an alternative PCR target used to screen environmental samples [97]. Table 1 presents the selected PCR primer pairs for these and other example marker genes, along with additional genes described in other molecular-based studies using tools other than PCR. Described below are many of the environmental parameters thought to drive *Vibrio* success, inferred from studies using a combination of culture-dependent and -independent techniques.

## 4. Predictors, Published Patterns, and Correlated Ecological Factors

Several water quality parameters may have strong influence on pathogenic *Vibrio* dynamics, partly due to their known optimal ranges. Temperature and salinity are often reported as drivers of *Vibrio* populations, regardless of geographical location of the waterbody. A variety of parameters are correlated with abundance and distribution in various regions and waterbody types and can be generally grouped into three categories: i) hydrological and/or meteorological factors, ii) abiotic factors that may be indicative of biological activity, and iii) purely biological factors. Additional considerations should be made when evaluating aquaculture studies, such as the now-known strains of *Vp* that are resistant to food pressure treatments [98,99]. An overview of environmental parameters potentially driving *Vibrio* success are provided here and in Table 2.

### 4.1. Oceanographic, Hydrographic, and Meteorological

#### 4.1.1. Temperature

Strong, positive correlations between water temperature and *Vv* distribution and/or abundance have been demonstrated in numerous studies, such as [46,55,110,114,120,123,140] among others. Using a combination of DP-CH and MPN-qPCR with primer pairs previously described [61], *Vibrio* concentrations were enumerated in surface waters, sediments, and oysters in coastal waters of two states in the subtropical northern Gulf of Mexico region [100]. Several environmental parameters, especially sea surface temperature (SST), contribute to *Vibrio* success in one or more of these substrates, along with chlorophyll *a* (chl *a*) and turbidity in the water column. Although many of the earlier studies finding increased *Vibrio* with higher sea surface temperatures were focused on subtropical or tropical locales, this strong positive relationship is also seen in additional, more temperate regions. For example, in North Carolina Atlantic east coast estuaries, DP/CH also confirmed strong relationships between temperature and *Vv* and *Vp*, but additional abiotic and biotic factors were also important considerations [12]. In the North Sea, where water temperatures were low and salinities high, infection occurrence was lower [141]. Warm waters were associated with high concentrations of *Vv* and *Vp* in surface waters of subtropical northern Gulf of Mexico but also in more temperate waters in the Pacific Northwest in the Chesapeake Bay [101].

There is little “correlation” of *Vibrio* with SST when short-term, within-season, evaluations are conducted in both culture-dependent [128] and culture-independent [134,142] types of studies. This is especially evident from studies where the entire sampling period occurs in the warm summer months [134] or from single-year evaluations where temperature is a strong factor [143]. Long-term, there are positive relationships between increased temperature with *Vibrio* in the zooplankton and phytoplankton seawater fraction [122], in the water column [112], and with *Vp* associated with shellfish [106]. However, in some cases, there may be a non-significant relationship of long-term effects of SST with *Vv* and *Vp* abundances in the water column [18]. Evaluation of epidemiological data and concomitant increases in sea surface temperatures [144,145,146] and other studies, reviewed by [147], are especially important for forming more confidence in ecological and environmental predictor models.

Increases in water temperature could also decrease water quality and induce favorable conditions for pathogen growth, such as salinity changes during droughts. For example, insubtropical surface waters in the northern Gulf of Mexico near Galveston, TX, USA contained the highest abundances of culturable *Vv* in waters with salinities of 7–16% and water temperatures between 30℃ and 37℃ [103,148]. Temperature was a driving factor in waters near Honolulu, HI, where *Vv* was low during the summer dry season, when waters were warm, and importantly, more saline [104]. In certain cases, droughts can promote distribution and abundance of pathogenic bacteria. However, some droughts do not seem to have much of an effect. For example, abundance of *Vv* in North Carolina estuaries was in fact reduced during a drought season compared to an above-average freshwater inflow event [149]. Similarly, a longer-term study (10 years) in the Neuse River Estuary showed that culturable *Vibrio* numbers were not necessarily associated with the three most commonly reported factors for predicting estuarine *Vibrio* abundance: salinity, temperature, and dissolved oxygen [150]. In a subtropical estuary, Charlotte Harbor of southwest Florida, there were no significant relationships between the presence of *Vv* and pH, salinity, turbidity, dissolved nutrients, or estuarine bacteria, an outlier from other studies [151]. These conflicting findings suggest that increased temperature coupled with a specific salinity range may provide more favorable *Vibrio* growth conditions.

#### 4.1.2. Salinity

Increases in salinity due to drought may actually decrease the levels of *Vv* associated with oysters, even when both culture-dependent and culture-independent methods of *Vibrio* measurement are employed [152]. Hindsight analyses indicate that *Vv* are more abundant in “wet” vs. “dry” years in the temperate Chesapeake Bay system [153]. In certain areas, such as subtropical estuaries, salinity can be a stronger determinant for *Vibrio* success than temperature. *Vibrio* are considered to be mesohalophiles (3–16 PSU). Thus, ecological surveys commonly report a negative relationship with salinity, especially at sites proximal to freshwater input [100,118,127,128]. Additional considerations should also include prevalence of the bacteria in surface waters as compared to bottom waters, where surface and bottom *Vv* may be differentially affected by temperature in the long term [154], a phenomenon likely present in any type of stratified estuarine environment. Some studies suggest that numbers increase proportionally with salinity. For example, a study near the Mississippi Gulf Coast reported a significant relationship between salinity (and turbidity) with *Vp* in surface waters and oysters [129]. *Vibrio* exposure can increase from exposure to low-salinity standing water caused by hurricane-induced flooding [155], but prior wind (see Section 4.1.4 and storm surge may also contribute to this problem. Systems affected by abundant freshwater influence, such as Indian tropical monsoonal estuaries, may also experience high *Vv* and *Vp* abundances, but in addition to salinity, other correlative factors such as eutrophication (see Section 4.2), reduced flushing leading to hypoxia, and wind direction reversal, are important [156].

Neither salinity nor SST alone is a strong predictor of pathogenic *Vv* and *Vp*, suggesting the need to consider other environmental factors influencing their distribution. Many studies of environmental samples consider the specific relationship between temperature and salinity to be the most primary consideration in *Vibrio* ecology. Warm waters with moderate-to-high salinity are typical for high abundances of *Vibrio* measured in both temperate and warmer locales [55,110]. The first study to detect and quantify co-occurring *Vibrio* species in West African coastal waters also confirmed that SST and salinity were driving environmental factors of *Vibrio* densities, and *Vv* was highest at the end of the wet season [124]. However, wind events (highest in this region’s dry season) were not considered. In myriad examinations among a variety of geographic regions and types of waterbodies, the one environmental parameter that is least predictive is salinity, where *Vibrio* abundances can be positively or negatively correlated with it, or where bimodal or non-linear models best fit the data (see “Relationship” column in Table 2).

#### 4.1.3. Alkalinity

Fluctuations in pH levels may also affect *Vibrio* ecology, particularly when preceded by heavy rainfall [157,158]. In Dauphin Island Bay, AL, USA, the optimal water pH for *Vv* and *Vp* were found to be ~7.0–9.0 [127]. In more temperate South Carolina, USA, *Vp* abundances were negatively correlated with pH, where HABs were prevalent in coastal stormwater detention ponds [115]. In a group of subtropical estuaries, pH was found to be correlated positively with *Vv*, but negatively with *Vp* [128]. Evidently, variation in driving factors can be highly region-specific and potentially seasonally driven, but tidal influence or freshwater inputs on fluctuating pH may also be the driving factor in *Vibrio* abundances. Tidal influence is an especially important predictive factor in estuaries, particularly shallow ones or those subjected to stratification [110,128,132].

#### 4.1.4. Wind

Importantly, besides SST, other considerations such as wind speed and air temperature should be included in ecological models. In a limited geographical area (tropical Florida, USA) there were relationships between coastal clinical cases and multiple meteorological factors [159]. One study elucidated the ecology and culturable *Vv* and *Vp* abundances in three substrates (surface waters, bottom sediments, and invertebrate biofilms) within the Pensacola Bay System, FL. High concentrations of *Vv* and *Vp* in surface waters may have been due to high wind speeds during the days leading up to sampling, which caused resuspension bacteria that would have otherwise been present within the sediments [128]. Additionally, wind-induced surface agitation may promote oxygenation, favoring *Vibrio* species whose oxygen preferences align with surface water conditions. For example, in the Chesapeake Bay, differential abundances of *Vv* and *Vp* were seen before and after an extreme storm event (Hurricane Irene), possibly because of increased turbidity from vertical mixing. Abundances in the water column, sediment, or both, were significantly correlated with several environmental parameters, notably secchi depth, total suspended solids, and tidal height [132]. Wind direction is also an important consideration, where the shape of the estuary dictates mixing depending on the vector [131], or where wind direction reversal, such as before or after significant storm events [156], can influence the water column *Vv* and/or *Vp* abundances.

#### 4.1.5. Turbidity

In coastal environments, an increased rate of fluvial input often increases turbidity, possibly driving *Vibrio* success [12,100,129]. Mixing and resuspension of bottom sediments transports sediment-dwelling *Vibrio* upward into surface waters, increasing their overall distribution and consequently reintroducing them to the water column. Not only does particle association allow *Vibrio* to evade grazing, particles are sources of nutrients and organic carbon [160,161]. In a recent study of coastal Alabama, USA, 30–50% of total *Vibrio* was associated with particles sized 5 μm or larger in a freshwater-driven system [131]. In a nearby northern Gulf of Mexico coastal region, light attenuation was, not unexpectedly, positively correlated with total suspended solids, but neither surface nor sedimentary *Vibrio* abundances were correlated with turbidity or light attenuation [128].

### 4.2. Abiotic Factors

#### Chlorophyll and Nutrients

Chlorophyll *a* (chl *a*) and nutrient concentrations are considered as abiotic factors, but they are indicative of biological activity within the ecosystem. Both biotic and abiotic factors generate selective pressures in ecosystems, which facilitate the emergence of pathogenic traits in the community [162]. In a northern temperate estuary, *Vp* associated with oysters were positively correlated with water column chl *a* [163], and the relationships of sedimentary and water column *Vibrio* with chl *a* were significant in several other regions [46,110,147]. Two regions near the subtropical Gulf of Mexico coast showed positive correlations of *Vv* and *Vp* with chl *a* in surface waters and sediments [100,128]. Eutrophication often promotes algal blooms. As blooms die and decompose, the dissolved and particulate organic matter released become fodder for heterotrophic bacteria such as *Vv* and *Vp*, which are efficient scavengers that often outcompete other non-pathogenic heterotrophs.

There is a clear connection between elevated nutrients and concentrations of *Vv* and *Vp*. In nutrient-rich detention ponds of South Carolina, USA [115], concentrations of a variety of nutrients—dissolved organic nitrogen, dissolved inorganic nitrogen, total dissolved phosphorus, and silicate—were important, where almost all correlated positively with both *Vv* and *Vp*. Similarly, in the Pensacola Bay System, surface water *Vp* positively correlated with total dissolved inorganic nitrogen and total Kjeldahl nitrogen [128]. Another study showed that *Vp* was moderately correlated with total dissolved phosphorus and total dissolved nitrogen in Rehoboth Bay, Delaware, USA [133]. Conversely, a strong inverse relationship was observed with ammonia for both *Vv* and *Vp* in North Carolina estuaries [12]. However, a predictive modelling survey in this region indicated that high dissolved organic carbon alone may not have a direct impact but can likely promote more rapid *Vibrio* growth when temperature and salinity ranges are viable [164].

### 4.3. Biological Factors

#### 4.3.1. Other Bacterial Pathogens

A positive correlation of *Vibrio* with total bacteria and/or coliforms is not unexpected, especially when *Vibrio* abundances are enumerated with culture-based methods [165]. Relationships between surface water *Vv* and fecal enterococci [124] was consistent with other earlier, culture-based studies [12,166]. Co-occurrences of *Vp* and other pathogenic bacteria were seen in a northern estuary with fecal contamination, concomitant with positive correlations with particulate matter and zooplankton abundances [167], again suggesting multivariate considerations must be made in predictive modeling. Indeed, in North Carolina estuaries, correlations of *Vv*, *Vp*, or both, were seen with several chemical, physical, and biological parameters, including salinity, turbidity, dissolved oxygen, concentrations of inorganic nutrients, and loads of other human pathogens such as *E. coli* and total coliform bacteria [12].

#### 4.3.2. Cyanobacterial and Eukaryotic Algal Blooms

The two different potentially pathogenic *Vibrio* species may respond differently to blooms following extreme storm events. In the Neuse River Estuary, North Carolina, USA, after an extreme storm event (Hurricane Matthew), several types of HABs were observed in surface waters, including cyanobacterial types and eukaryotic dinoflagellates, both measured using deep sequencing of the *hsp60* gene [136]. Although *Vibrio* abundances were estimated via relative abundances in deep sequencing, significant correlations with the typical “environmental factors” were seen, but differing relationships with HABs, depending on the type of HAB and the *Vibrio* species, were also observed. Correlations with a cyanobacterial bloom with *Vp* and *Vv* were positive and negative, respectively, but both *Vibrio* species were positively correlated with dinoflagellate blooms. Therefore, after extreme storm events, water quality, including various photosynthetic pigments such as chl *a* (as a proxy for potential HABs) should therefore be monitored closely when assessing aquaculture risks. These risks were also assessed in coastal Mediterranean, in a fish aquaculture area, where several water quality markers were correlated with multiple *Vibrio* species, based on 16S rRNA profiles and culturable bacteria counts [168]. Interestingly, but not surprisingly, bacterial species often associated with fish lesions and/or mortality were associated with relatively high levels of the human pathogens in the *Vv* and *Vp* species markers found in the deep sequencing amplicon data.

Species within the copiotrophic *Vibrio* genus are often associated with surfaces, which could include association with harmful algal bloom (HAB)-forming cyanobacteria, eukaryotic phytoplankton and zooplankton. Indeed, a combination of higher SST and chitinous plankton abundance are associated with higher *Vv* [143], and distinct pathogenic types are associated with distinct groups of chitinous eukaryotic plankton and follow particular patterns with respect to environmental conditions such as salinity and temperature [39]. Capabilities of extracellular polysaccharide matrix formation for *Vibrio* and their resulting increased fitness were previously reviewed elsewhere [160,169]. Archived plankton samples may be screened with molecular methods to further understand long-term SST increases and prediction of *Vibrio* dynamics. Using genus-specific PCR primers, *Vibrio* associated with plankton were positively correlated with SST, in addition to other climate indices, namely, the Northern Hemisphere Temperature and the Atlantic Multidecadal Oscillation [122].

Co-occurrences of high relative abundances of *Vibrio* and HAB species are also of particular interest. In temperate coastal and inland waters of Delaware, USA, higher abundances of *Vibrio* are concomitant with various eukaryotic species, often associated with blooms, as assessed with both size fractionation and in whole water analyses [44,133]. The early Delaware study relied solely on phylogenetic markers to identify relative abundances of *Vibrio* species of interest, but the 2022 Rosales study in the same geographic region used more specific genetic markers that are most likely to reflect the more pathogenic strains of *Vv* and *Vp*. Additionally, in subtropical estuaries of the Gulf of Mexico coast (Alabama USA), the findings are similar to those of temperate Delaware [131]. Size fractionation prior to DNA extraction and analysis using qPCR with the most reliable marker genes indicate that *Vv* and *Vp* are associated with blooms of *Akashiwo sanguinea* and *Heterocapsa* spp. Even more notably, and supporting other findings [136], the effect of wind prior to the HAB bloom influences the strength of the responding *Vibrio* increased abundances.

In both northern and southern temperate regions, as well as subtropical waters, abundances are positively correlated with higher relative amounts of markers of algal or cyanobacterial biomass, including but not limited to photosynthetic pigments, microscopy counts, HAB marker genes, or relative HAB abundances based on phylogenetic marker gene datasets [131,137]. Geographic considerations, including the maximum or persistence of SST must also be included in predictive modeling. In a *Prorocentrum* bloom of temperate coastal surface seawater (East China Sea), based on *Vibrio*-specific genes in a metagenomics dataset, as well as 16S rRNA relative abundances in amplicon deep sequencing data, there was a strong (but not statistically significant) association between *Vibrio* genera gene abundances and HAB algal biomass [135]. In contrast to findings of coastal South Carolina, USA, tidal creeks and nearby retention waterbodies, where increases in either or both *Vv* and *Vp* were associated with harmful cyanobacterial blooms and harmful eukaryotic algal blooms [115], there were no such associations in a partner study in Puget Sound of Washington state, USA, a more temperate environment [170]. Notably, correlations of *Vp* with HABs are more pronounced with higher SST. In the Puget Sound, there were clear blooms of three harmful species during the study period (*Pseudo-nitzschia*, *Alexandrium* and *Dinophysis*), but these events did not coincide with increases in *Vp* [170]. Also in this study, the only water column nutrient significantly associated with *Vp* was silicate, unlike the findings of many other studies suggesting strong correlations between *Vv* and *Vp* abundances with nitrogen- and phosphorous-containing nutrients (see Section 4.2).

The known association of *Vv* and *Vp* in the “particle-associated” fraction (often meaning associated with eukaryotic plankton) also emphasizes the need for size fractionation of water column samples prior to screening for the bacteria. This type of size fractionation is rare, but is becoming more common in environmental studies such as [131], resulting in insightful conclusions about the *Vibrio* ecology considerations when planning and modeling to improve aquaculture safety. Indeed, the composition of the plankton community may dictate the relative success of *Vv* and *Vp* [131] or bacteria in the *Vibrio* genus [143]. Based on a study using only culture-independent methods, relative abundances of planktonic *Vibrio* are found to be higher in the FL fraction as compared to PA (>3 micron), but the qPCR primers used were only able to enumerate *Vibrio* at the genus level [142].

#### 4.3.3. Seagrass Density

Coastal areas with high water movement can facilitate *Vibrio* transport and distribution. As recently examined with culture-dependent tools [128], it has long been known that *Vibrio* are enriched in sediments, where their abundances can exceed water counts by a log or more, and temporal fluctuations of *Vv* and *Vp* abundances in sediments are pronounced and tied closely to SST [171]. Wind intensity, changes in direction, and changes in wind patterns may mix sedimentary *Vibrio* into the water column, increasing chances of human encounters or bringing more into contact with oyster and other shellfish. Fortunately, it is possible that these effects may be mitigated by submerged aquatic vegetation that stabilize sediments. In temperate eelgrass (*Zostera marina*) meadows, all *Vibrio* and *Vv* abundances were reduced, based on culture-dependent measurements [138].

Similarly, a negative correlation of *Vibrio* plus other pathogens was seen with other types of seagrasses in tropical coasts of Indonesia based on deep sequencing of the 16S rRNA gene [139]. Unfortunately, inferring the relationship of grass density with strain- and species-specific *Vibrio* abundances is not possible with this method, and to date, the number of seagrass density studies is limited. Additionally, it has recently been suggested that surfaces of macroalgae or seagrasses could harbor higher *Vibrio* densities than in environs absent of macrophytes [172]. The promise of decreased pathogen loads in waterbodies with high-density seagrass beds, however, is likely due to the sediment stabilization by healthy seagrasses.

#### 4.3.4. Other Anthropogenic Factors

In addition to all the environmental factors described above, introduction of microplastics and heavy metals into waterbodies near aquaculture facilities may also increase abundances. For example, *Vp* enumerated with *trh* qPCR was positively correlated with heavy metals in water samples, but the impact of metals was also seen for general bacteria, *E. coli*, *Pseudomonas aeruginosa*, among other groups [134]. Additionally, growth and biofilm formation by both pathogenic and non-pathogenic types of *Vv* are positively correlated with increasing iron concentrations [173]. Introduction of heavy metals, antibiotics, or both, may affect the rate or mechanism ofARG transfer on transposable elements [174]. There are several recent general reviews on antibiotic resistance genes in *Vp* and *Vv*, including [81,175,176].

Extracellular polysaccharide matrix formation by *Vv* and *Vp* leads to fitness on surfaces, both sessile and planktonic [160,169]. Therefore, it is not surprising that microplastics loads in the water column are a risk factor for enhanced *Vibrio* abundances. *Vibrio* bacteria are also enriched in the plastisphere. A survey from the North and Baltic Seas reported that *Vibrio*-attached microplastics tend to be more brittle, perhaps due to degradation by high winds and storm events [177]. These degraded microplastics may increase surface area and create greater attachment sites for these particle-associated microbes. More general molecular tools for enumerating bacterial families, including *Vibrioaceae*, were previously used in the context of fluorescence *in situ* hybridization (FISH) to follow succession of colonizers on plastic surfaces; *Vibrio* are important in early biofilm colonization [178].

In northern estuarine waters, *Vibrio* have different responses to various artificially introduced substrates, where wood or plastics result in the expression of different groups of genes, resulting in different proteomes of the different substrates’ bacteria [179]. Colonizing sargassum and microplastics in subtropical waters of the Caribbean and Sargasso Seas all apparently rely on the expression of the pilus-related MSHA operon, including the *mshA* gene. The MSHA is necessary for adhesion and biofilm formation, regardless of substrate [180]. As extensively reviewed [181], sessile *Vibrio* abundances on these artificial surfaces are similar to those in naturally occurring substrates, both in fixed environments as well as on planktonic surfaces such as floating particles or larger plankton.

#### 4.3.5. All Factors

Historically, several basic water quality parameters such as temperature and salinity were thought to have strong influence on pathogenic *Vibrio* dynamics, partly due to their known optimal ranges. As described in detail in Section 4.1 through Section 4.3, myriad examples exist demonstrating a combination of factors, both biotic and abiotic, that correlate very well with increased *Vv* and *Vp* abundances. Wind-mixing of sediments, seagrass bed densities, and other climate-related factors can influence both *Vibrio* distribution, as well as the distribution of other planktonic groups such as zooplankton and phytoplankton, now known to be the biotic predictors of *Vibrio* blooms. Importantly, it is now clear that multivariate analyses are needed to predict *Vv* and *Vp* abundances’ correlations with meterological, biological and abiotic environmental factors [39,182], since oceanographic and hydrographic perturbations affect all trophic levels, including those to which *Vibrio* may be attached. In addition, interpretation of correlative relationships should also consider the enumerating tools used, which are overviewed in Section 5 (below).

## 5. Future Outlooks, Advancements, and New Molecular Tools

Ecological or environmental parameter predictions, to date, are based on various enumeration methods, each of which has its own set of limitations. Limitations of enumeration methods could include sensitivity, PCR primer bias, or underestimations due to “cultivability” of the environmental strains of interest. Common to many types of naturally occurring bacteria, the “Viable but Not-Culturable” (VBNC) state can confound models. Bacteria in the VBNC state may cause us to miss those “overwintering” *Vibrio* from sediments, or *Vibrio* associated with oysters, when using culture-based methods [32,183]. The in situ water temperature also dictates culture growth rates, in particular environmental *Vv* from cooler samples [184]. Although the VBNC state, metabolic survival and pathogenesis responses of these bacteria have been extensively reviewed elsewhere [26,185,186,187], most current environmental monitoring types of studies still do not consider the impact of inclusion of culture-based methods on *Vibrio* abundance underestimates. Modeling based on data from most studies using partially culture-based methods described in Table 2, such as liquid culture-based (MPN-) or solid medium (DP-) culturing are at risk of underestimating *Vv* and *Vp* that are possibly pathogenic but resist growth on our current culture media.

Abundances gleaned from metagenomics studies of samples associated with enteric pathogens can be as reliable as, or at least consistent with, culture-based methods. For screening environmental samples, tying strain taxonomy with important functional genes such as individual ARGs or multi-drug resistance transporters is possible with metagenomics [188]. Other more specialized types of molecular-only types of screening tools are available, such as LAMP-PCR-based assays for *Vp*, can be used, but these methods are limiting due to cost, extensive pretreatment needs, or other problems [189]. Care should be taken with interpretations of studies using tools enumerating *Vibrio* at the genus level, or even at the species level without considering relative species abundances of pathogenic and non-pathogenic species [18].

Sensitivity of current assays, both those that include a culture-dependent step, as well as those that bypass enrichments, is generally not a problem. For example, methods in DP studies or others that use an enrichment step prior to PCR/qPCR, all have very low detection limits, ranging from ~<1 [21] to 5–15 colony forming unit (CFU) per mL of seawater [54,128]. From oysters, DP methods can detect as low as 5–10 CFU per gram of oyster homogenate sample [121,190]. Multi-step assays that rely on liquid culture-based enrichment steps first, prior to PCR or qPCR, are even more sensitive, since the enrichment allows for growth prior to detection. Ranges of sensitivity for these types of studies are quite variable. For example, in a variety of substrates from both subtropical and temperate locales, the lower detection limit of either *Vv* or *Vp* in all substrates tested was 1 to >80,000 CFU per unit of substrate tested, specifically 1–250 CFU/mL (water), 10–25,000 CFU/g (oyster), and 100 to 83,333 CFU/g (sediment) [101].

PCR primer bias could also lead to underestimation of true *Vibrio* abundances. For example, most of the recent studies employing PCR-based verification use PCR primers designed and published over a decade ago (“Gene” column in Table 2). One exception is a recent study that developed newer functional gene assays, including for one for *sodB* and related genes, which may assist *Vv* in transitioning from the environment into a virulent state associated with its host [74]. PCR primer biases may result from direct PCR (or methods that use a culture-based step first followed by PCR or qPCR), since primer pairs (Table 1) may not match well to all representative strains’ sequences. PCR primer bias does not create an issue with sensitivity, rather the community composition of pathogenic strains of *Vibrio* may not be accurately represented with use of outdated primer sequences. To remedy this, continual development of qPCR primer pairs should be carried out, refining them to be inclusive of more recently submitted sequences from known pathogenic strains.

However, there is hope; examples of more efficient direct analyses of DNA are becoming more prevalent, such as direct qPCR [191], and metagenomics [192], where both of these types of methods by-pass the culture-dependent step completely. Sensitivity of qPCR-only studies is, like with methods containing a culture step, very good. For example, *Vp* detection is as low as 1 to 10 copies per mL of seawater with *trh* primers [134]; with these and other primers for *Vp* and for other *Vibrio* species, a similar range is detectable with the very sensitive qPCR method [137]. Detection of *Vv* with direct qPCR is also very sensitive; cell equivalents in DNA enumerated with *vvhA* qPCR is as low as 15 CFU per mg of homogenized fish tissue [54]. PCR primer bias can be overcome with the use of PCR-independent detection of functional genes in genome segments of the entire microbial consortia. For instance, in metagenomics, sensitivity may be a problem, but the benefits gleaned (new sequences of functional genes obtained without a PCR step) may lead to improved PCR and qPCR primer pairs [192].

Increases in vibriosis associated with finfish aquaculture has been reviewed extensively elsewhere [193]. A recent systematic review of foodborne-*Vibrio* evaluated temperature and other non-environmental parameters provides insights into predictions of *Vibrio* distributions in finfish aquaculture [194]. Although specific serovars can affect the health of aquaculture species such as eels and other teleosts [195], the scope of this review is limited to human pathogens among *Vv* and *Vp*, and thus, particular attention is paid to increasing *Vibrio* abundances in shellfish [190]. Oysters, often eaten raw, are well-known to harbor high abundances of both *Vv* and *Vp*, in their liquor, on their shells, and/or in the meat. Seafood-borne infections are dominated by these two species as well as *V. cholerae* [64]. Oyster-derived *Vp* outbreaks in northern US environments on both Atlantic and Pacific coasts and in the Gulf of Mexico region [196], as well as *Vv* outbreaks in a variety of locales [197] indicate the pervasive nature of these naturally occurring species.

Numerous earlier, and all subsequent, studies (too many to list, but many of which are listed with “Oyster” in the Substrate column of Table 2) emphasize the need for good screening methods to improve aquaculture efforts. Importantly, there may be differential influences in the changes in salinity, TSS, or even water depth on the loads of pathogenic *Vv* in oysters and clams, as well as an increased chance of finding pathogenic *Vp* in cooler, rather than warmer, waters [38]. These studies have demonstrated how using genomics and molecular methods to call for and/or enumerate these genes can distinguish virulent strains of *Vibrio* in environmental samples.

Genes of interest for many studies focused on those in pathogenic *Vv* and *Vp*, including *tdh*, *trh*, and *vvhA.* A 1999 study successfully defined and developed a 5′ nuclease probe-based qPCR method for quantifying *Vp* in oyster tissue homogenate [59]. When compared to positive detection in pure culture strains, *trh*, but not *tdh*, was detected in one strain, suggesting that assaying for both genes may be necessary to detect all hemolysin-producing strains. Some published primer pairs also exist for detection and enumeration of *Vv* with qPCR forgoing the use of probes [56]. Such a design relies solely on SYBR Green I fluorescence in qPCR for the detection of amplicon production during the course of the PCR, a more cost-effective approach than probe-dependent assays. Additionally, careful PCR primer design may improve detection since TaqMan probes add a layer of specificity, reducing the ability to include degeneracy codes to better capture the full diversity of pathogenic strains’ gene sequences [198].

Pathogenicity-associated target genes may appear to be present in species other than *Vp*. For example, *trh*, a target gene that was only reported to be present in *Vp* [199] was found in what was later identified as *V. alginolyticus*, a near-neighbor species, suggesting a positive PCR result could be associated with presence of pathogenic *Vp* [200]. The lineage of a newly evolved pathogenic *Vp* strain expanded its distribution along the North American Atlantic coast. It was dubbed as a new “sequence type”, number 631 (ST631) exhibiting a similar virulence gene profile to members of a related ST36 clade, a group of *Vp* known to contain both *tdh* and *trh* [201]. Methods of detecting such highly adaptable pathogens need to be attuned often to accommodate the effects of a rapidly changing environment.

Increasing global temperatures due to climate change portends an increase in frequency and/or intensity of heavy rainstorms, which could promote *Vibrio* abundance. However, there are now better predictive tools such as machine learning modeling of hydrodynamic factors influencing particle movement [202], along with multivariate analyses to predict *Vv* and *Vp* abundances’ correlations with meteorological, biological and abiotic environmental factors [39,182]. There is promise for improved predictive models without sampling oyster tissues; this is especially true in aquaculture environments with low tidal range. Recently described in a study of water column and oyster abundances, the predictive patterns with fluctuations in environmental parameters may be determined by water column abundances alone [110], potentially allowing for time and cost-savings. Finally, the knowledge that healthy densities of submerged aquatic vegetative material can possibly mitigate some of these problems is a good start; further studies on seagrass beds and associated water column, sediment and shellfish-associated *Vibrio* are needed. These improved models and additional information on the potential for seagrass mitigation may assist local, regional, and other policy-makers in planning improvements for aquaculture facilities and human recreation. Methodological shifts in measurement approaches, away from those that depend on culturing as a first step, and towards evaluation of DNA or RNA extracted directly from the substrate will also allow for more rapid detection, accurate quantification, and a more comprehensive characterization of the types of bacteria present.

## Figures and Tables

**Table 2 microorganisms-11-02502-t002:** Known environmental factors associated with changes in *Vv* and *Vp* abundances. Studies listed show strong positive or negative correlations of *Vibrio* species with factors listed (Env. Column). Method: tools used to enumerate *Vv* or *Vp* are indicated (if PCR and/or qPCR primers were used, the original reference for the primer sequences is provided). Quantification methods for studies examining correlating environmental factors included: MPN (most probable number), DP (direct plating), CH (colony hybridization), PCR (polymerase chain reaction) or qPCR (quantitative PCR), BOX-A1R-based repetitive extragenic palindromic-PCR (BOX-PCR), or simply “culture”, indicating liquid culture methods. CHROMAgar™ is a specialized direct plating method and is listed separately from DP. Some studies used only metagenomics and/or deep sequencing of amplicon libraries of a single gene, where the genes were either 16S rRNA or other phylogenetic markers. Genes specific to *Vv* and/or *Vp* are listed in Table 1. Positive, negative, or other non-linear relationships of environmental factors with either or both *Vv* and *Vp* species are indicated, where a check mark indicates the effected *Vibrio* species, and a check in parentheses indicates that species-level confirmation is not possible due to the limitations of the enumerating tool used.

Study	Asso. ^1^	Env. ^2^	*Vv*	*Vp*	Substrate	Method ^3^	Gene(s) ^4^
*Oceanographic*, *hydrographic*, *meteorological*							
[12]	Positive	*Temperature*	✓	✓	Surface waters	DP/CH, conf. PCR	*tlh* [59]
	Positive		✓	✓	Sediments		
[100]	Positive		✓	✓	Surface waters	DP/CH, MPN-qPCR	*tdh*, *trh*, *tlh* [61]
[101]	Positive		✓	✓	Surface waters	DP/CH, MPN-qPCR	*tdh*, *trh*, *tlh* [61] *vvhA* [102]
	Positive		✓	✓	Sediments		
	Positive		✓		Oysters		
[103]	Negative			✓	Oysters	DP/CH, conf. PCR	*tdh* [59]
[104]	Positive		✓		Surface waters	DP, qPCR	*vcgC* [105]; *vvhA* [34]
[55]	Positive		✓		Surface waters	MPN-qPCR	*vvhA* [55]
[106]	Positive			✓	Oysters	MPN-PCR	*tdh* [107]; *trh* [108]
[109]	Negative			✓	Surface waters	MPN-PCR	*tlh*, *trh*, *tdh* [59]
[97]	Positive			✓	Surface waters	MPN-PCR	*tlh*, *tdh*, *trh*, *toxR*, *vpm* [61,65]
	Positive			✓	Oysters		
[110]	Positive		✓	✓	Surface waters	MPN-qPCR	*vvhA* [34]; *pilF* [111]; *trh*, *tdh*, *tlh* [61]
[112]	Positive		✓	✓	Surface waters	DP/CH, conf. PCR	*tdh* [102]; *trh* [61]; *vvhA* [113]
[114]	Positive			✓	Oysters	N/A; meta-analysis of oyster surveys and previous publications	
[115]	Positive		✓		Surface waters	DP/CH, conf. PCR	*tdh/trh* [61]; *vvhA* [116]
[46]	Positive		✓		Surface waters	MPN-qPCR	23S rRNA gene nested with *Vv*-specific 23S [117]
[118]	Positive			✓	Surface waters	BOX-PCR	*ToxR* [119]
[120]	Positive		✓		Surface waters	qPCR, culture	*vvhA* [34]
	Positive		✓		Oysters		
	Positive		✓		Vegetation		
[121]	Positive			✓	Oysters	DP/CH, MPN-qPCR	*tdh*, *trh*, *tlh* [61]
[122]	Positive		(✓)	(✓)	Plankton fraction (phytoplankton and zooplankton)	qPCR	*Vibrio* genus 16S rRNA gene [45]
[123]	Positive		✓		Surface waters	DP/PCR, qPCR	*vvhA* [116]
[124]	Negative	*Moderate to low salinities*	✓	✓	Surface waters	MPN-qPCR	*tdh*, *trh* [59]; *vvhA* [125]
[104]	Nonlinear		✓		Surface waters	MPN-qPCR	*vcgC* [105]
[110]	Negative		✓		Surface waters	MPN-qPCR	*vvhA* [34]
	Negative		✓		Oysters		
[55]	Negative		✓		Surface waters	MPN-qPCR	*vvhA* [55]
[110]	Convex		✓	✓	Oysters	MPN, qPCR	*tdh*, *trh*, *tlh* [61]; *vvhA* [34]
[100]	bimodal		✓		Surface waters	DP/CH, MPN-qPCR	*vvhA* [102]; *tdh*, *trh*, *tlh* [61]
				✓			
				✓			
				✓	Oysters		*tdh*, *trh*, *tlh* [61]
				✓			
				✓	Sediments		*tdh*, *trh*, *tlh* [61]
			✓				
[126]	Positive			✓	Shellfish	DP/CH	*tdh*, *trh* [59]
[112]	Nonlinear		✓	✓	Surface waters	DP/CH, conf. PCR	*tdh* [102]; *trh* [61]; *vvhA* [113]
[115]	Negative		✓	✓	Surface waters	DP/CH, conf. PCR	*tdh*, *trh*, *tlh* [61]; *vvhA* [116]
[46]	Positive		✓		Surface waters	MPN-qPCR	23S rRNA gene nested with *Vv*-specific 23S [117]
[127]	Negative		✓	✓	Fish intestines	qPCR, DP-CH	*Vibrio*-specific 16S rRNA [45]
[103]	Bimodal			✓	Oysters	DP/CH, conf. PCR	*tdh* [59]
[118]	Negative			✓	Surface waters	BOX-PCR	*ToxR* [119]
[120]	Positive		✓		Water	qPCR, culture	*vvhA* [34]
	Positive		✓		Sediments		
[128]	Negative			✓	Surface waters	DP, conf. PCR	*tdh*, *trh*, *tlh* [61]
[123]	Positive		✓		Surface waters	DP/PCR, qPCR	*vvhA* [116]
[100]	Positive			✓	Surface waters	DP/CH, MPN-qPCR	*tdh*, *trh*, *tlh* [61]
	Positive			✓	Oysters		
	Positive			✓			
	Positive			✓	Sediments		
	Positive			✓			
	Positive			✓			
[110]	Negative		✓		Surface waters	MPN-qPCR	*pilF* [111]; *vvhA* [34]
[129]	Positive			✓	Surface waters	DP/CH, MPN-qPCR	*tdh*, *trh*, *tlh* [61]
	Positive			✓	Oysters		
[130]	Negative			✓	Seals	DP/CH, conf. PCR	*tdh*, *trh* [59]
[121]	Positive			✓	Oysters	DP/CH, MPN-qPCR	*tdh* [61]
[123]	Positive		✓		Surface waters	DP/PCR, qPCR	*vvhA* [116]
[128]	Positive	*7.0 > pH < 9.0* ^ 5^	✓		Bottom waters	DP, conf. PCR	*vvhA* [116]; *tdh*, *trh*, *tlh* [61]
	Negative			✓	Surface, bottom waters		
[115]	Negative			✓	Surface waters	DP/CH, conf. PCR	*tdh*, *trh*, *tlh* [61]
[131]	Negative		✓	✓	Surface waters	MPN-qPCR	*tdh*, *trh*, *tlh* [61]
[128]	Positive	*Tidal influence* ^ 6^	✓		Surface waters	DP/CH, conf. PCR	*vvhA* [116]; *tdh*, *trh*, *tlh* [61]
	Positive		✓		Sediments		
[132]	Negative	*Turbulence* ^ 7^	✓		Sediments	MPN-PCR	*vcgC* [105]; *tdh*, *trh*, *tlh* [61]
	Negative			✓			
	Negative			✓	Surface waters		
[110]	Negative		✓		Oysters	MPN-qPCR	*pilF* [111]; *vvhA* [34]
[128]	Positive		✓	✓	Surface waters	DP, conf. PCR	*vvhA* [116]
[131]	Negative		✓	✓	Surface waters	MPN-qPCR	*vvhA* [34]; *vvhA* [34]; *tdh*, *trh*, *tlh* [61]
[129]	Positive	*Turbidity*		✓	Surface waters	DP/CH, MPN-qPCR	*tdh*, *trh*, *tlh* [61]
	Positive			✓	Oysters		
[131]	Positive		✓	✓	Surface waters	MPN-qPCR	*vvhA* [34]; *tdh*, *trh*, *tlh* [61]
[129]	Positive			✓	Surface waters	DP/CH, MPN-qPCR	*tdh*, *trh*, *tlh* [61]
	Positive			✓	Oysters		
*Abiotic factors that may be indicative of biological factors*							
[100]	Positive	*Chlorophyll*	✓	✓	Surface waters	MPN-qPCR	*tdh*, *trh*, *tlh* [61]
	Positive		✓	✓	Sediments		
[110]	Positive			✓	Surface waters	MPN-qPCR	*tdh*, *trh*, *tlh* [61]
[112]	Positive		✓	✓	Surface waters	DP/CH, conf. PCR	*tdh* [102]; *trh* [61]; *vvhA* [113]
*Nutrients* ^ 8^							
[133]	Positive	*TDN*		✓	Surface waters	MPN-PCR	for *Vv*, *vvhA* [116], and *vcg* [105]; for *Vp*, *tdh*, *trh*, *tlh* [61]
[115]	Positive	*TDP*		✓	Surface waters	DP/CH, conf. PCR	*tdh*, *trh* [61]; *vvhA* [116]
	Positive	*TDP*		✓			
	Positive	*PO* _4_ ^−^		✓			
	Positive	*DIN*	✓				
	Positive	*PO* _4_ ^−^	✓				
	Positive	*Si*	✓	✓			
	Positive	*TDN*	✓	✓			
	Positive	*DON*	✓	✓			
	Positive	*TP*	✓	✓			
	Positive	*TDP*	✓	✓			
[118]	Negative	*TN*		✓	Surface waters	BOX-PCR	*ToxR* [119]
	Negative	*TP*		✓			
[128]	Positive	*DIN*		✓	Surface waters	DP, conf. PCR	*tdh*, *trh*, *tlh* [61]
	Positive	*TKN*		✓			
[134]	Negative	*PO* _4_ ^3−^		✓	Surface waters	qPCR	*tdh*, *trh* [61]
	Negative	*NO* _3_ ^−^		✓			
	Negative	*NO* _2_ ^−^		✓			
	Negative	*NH* _4_ ^+^		✓			
*Biological factors*							
[124]	Positive	*Vibrio cholerae*	✓	✓	Surface waters	MPN-PCR	*tdh*, *trh* [59]; *vvhA* [125]
[12]	Positive			✓	Surface waters	DP/CH, conf. PCR	*tdh*, *trh* [59]
[124]	Positive	*FIB* ^ 9^		✓	Surface waters	MPN-PCR	*tdh*, *trh* [59]
	Positive		✓		Surface waters		*vvhA* [125]
[12]	Positive		✓	✓	Surface waters	DP/CH, conf. PCR	*tdh*, *trh* [59]
[74]	Positive		✓		Surface waters	qPCR	*vvhA* [34]
	Positive		✓		Mesocosm	RT-qPCR	*sodB* [74]
*Other organisms* ^ 10^							
[115]	Positive	Multi-species harmful cyanobacteria bloom ^11^	✓	✓	Surface waters	DP/CH, conf. PCR	*tdh*, *trh*, *tlh* [61]; *vvhA* [116]
	Positive	Multi-species HAB	✓				
	Positive	*H. rotundata*, non-HAB euglenophyte *Eutreptiella* spp. ^12^		✓			
[44]	Positive	*Heterosigma* spp., total dinoflagellates	(✓)	(✓)	Surface waters	qPCR	*rpoA* [43]
[133]	Positive	*H. akashiwo*, *G. instriatum*^ 13^		✓	Surface waters	MPN-PCR	*tdh*, *trh*, *tlh* [61]
			✓				*vvhA* [116]; *vcgC* [105]
[135]	Positive	*Prorocentrum*	(✓)	(✓)	Surface waters	Metagenomics, 16S deep seq.	Phylogenetic markers, full genome
[136]	Positive	Dinoflagellates ^14^	(✓)	(✓)	Seawater	*hsp60*, 16S deep seq.	Phylogenetic markers
	Negative	Cyanobacterial bloom ^14^	(✓)				
	Positive			(✓)			
[131]	Positive	*A. sanguinea*, *Heterocapsa* spp. ^15^	✓		Surface waters	MPN-qPCR	*vvhA* [34]; *tdh*, *trh*, *tlh* [61]
[137]	Positive	Diatom-dominated phytoplanktonbloom		✓	Surface waters	qPCR	*tdh*, *trh*, *tlh* [61]
[138]	Negative	*Seagrass* Δ ^16^	(✓)	(✓)	Surface waters	CHROMagar	N/A
[139]	Negative		(✓)	(✓)		16S deep seq.	Phylogenetic markers

^1^ Association, or relationship, with indicated environmental factor. ^2^ Correlated environmental factor. ^3^ Quantification method used; Conf. PCR: confirmed with PCR; deep seq.: deep sequencing. ^4^ Gene(s) detected with probes or amplified *via* PCR or qPCR. Primer pair or probe name and reference provided. ^5^ Reported pH conditions between 7.0 and 9.0. ^6^ Factors directly associated with tidal influence, including tidal range, tidal coefficient, or salinity stratification. ^7^ External physical factors related to resuspension of the water column, including wind direction, wind speeds, and storm events. ^8^ Nutrient concentrations. TDN: total dissolved nitrogen. TDP: total dissolved phosphorus. DIN: dissolved inorganic nitrogen. Si: silicate. DON: dissolved organic nitrogen. TP: total phosphorus. TN: total nitrogen. TKN: total Kjeldahl nitrogen. ^9^ Fecal indicator bacteria (fecal enterococci, total coliforms). ^10^ Other microorganisms including cyanobacterial harmful algal blooms (HABs), eukaryotic phytoplankton HABs, zooplankton. ^11^ Species include *Cylindrospermopsis raciborskii*, *Anabaena* spp. ^12^
*Heterocapsa rotundata.*
^13^
*Heterosigma akashiwo*, *Gyrodinium instriatum.*
^14^ following extreme storm event. ^15^
*Akashiwo sanguinea.*
^16^ Change in seagrass density.

## Data Availability

Not applicable.

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
