# Peer review of "Tools to Enumerate and Predict Distribution Patterns of Environmental Vibrio vulnificus and Vibrio parahaemolyticus"

_microorganisms, 2023, doi:10.3390/microorganisms11102502_

Round 1
Reviewer 1 Report
This review systematically introduced the detection methods of Vibrio vulnificus and Vibrio parahaemolyticus, and summarized the ecological driving factors for the occurrence of these two types of Vibrio(Vv and Vp). Based on the development of existing technology, specific suggestions for tool improvement are proposed to address the occurrence of Vibrio disease. The strengthening of the warning model proposed in this review has a certain promoting effect on the prevention and control methods of Vibrio disease. However, among the factors that cause changes in the abundance of Vibrio, most of them are highly correlated with climate. Perhaps we should focus on exceptional events beyond this, and provide a more systematic explanation of the relevant influencing factors, which can more fully describe the key reasons related to Vibrio outbreaks.
And there just some miner questions in the manuscript:
1)In the line 186, “encoded by toxR in V. cholerae”, does it mean that toxR originated from Vibrio cholerae or was it first discovered on Vibrio cholerae?
Reviewer 2 Report
The manuscript submitted by Waidner summarized the tools to enumerate and predict distribution patterns of environmental Vibrio vulnificus and Vibrio parahaemolyticus. The topic is interesting, but I think the manuscript should undergo a significant revision. The following are my comments:
1. Line 25, the authors stated that “Three major human pathogens in the genus include Vibrio vulnificus, V. parahaemolyticus, and V. cholerae, the first two of which are considered in this review.” In my opinion, the authors should give objective reason for that the V. cholerae was excluded.
2. Line 80, the authors mentioned “VBNC”, but there is another state “persister” for bacteria in the environment, as a review, I think this specific state should also be mentioned.
3. From line 88, the “new” molecular tools for the detection of Vv and Vp were mentioned. In my opinion, this part has no summarized description, but only the related studies were listed one by one. Also, as new detection methods, the detection limit and linear range had better be given, and the superiority over the “old” ones should be clearly listed.
4. In the section of 4.1, the description is slightly superficial, in-depth interpretation should be given to interpret why or how the oceanographic, hydrographic, and meteorological factors drives the distribution of Vv and Vp.
I have no comments on English writing, but the manuscript should be checked carefully for the writing of the gene names. The gene names should be italic.
Round 2
Reviewer 2 Report
The authors have make well improvement to the manuscript, and I have no further comments now.